# 11β-HSD1 Inhibitor Alleviates Non-Alcoholic Fatty Liver Disease by Activating the AMPK/SIRT1 Signaling Pathway

**DOI:** 10.3390/nu14112358

**Published:** 2022-06-06

**Authors:** Ying Chen, Jiali Li, Meng Zhang, Wei Yang, Wenqi Qin, Qinzhou Zheng, Yanhui Chu, Yan Wu, Dan Wu, Xiaohuan Yuan

**Affiliations:** Heilongjiang Key Laboratory of Tissue Damage and Repair, College of Life Science, Mudanjiang Medical University, Mudanjiang 157011, China; chenying9435@126.com (Y.C.); lijiali0530@163.com (J.L.); zhangmengzi1125@163.com (M.Z.); yangwei19970531@163.com (W.Y.); qinwenqi0717@163.com (W.Q.); zz804873737@163.com (Q.Z.); yanhui_chu@sina.com (Y.C.); wuyan@mdjmu.edu.cn (Y.W.); wudan@mdjmu.edu.cn (D.W.)

**Keywords:** 11-beta-hydroxysteroid dehydrogenase type 1 (11β-HSD1), curcumin, non-alcoholic fatty liver disease (NAFLD), lipid metabolism, anti-inflammatory

## Abstract

We investigated the effect of an 11β-HSD1 inhibitor (H8) on hepatic steatosis and its mechanism of action. Although H8, a curcumin derivative, has been shown to alleviate insulin resistance, its effect on non-alcoholic fatty liver disease (NAFLD) remains unknown. Rats were fed a high-fat diet (HFD) for 8 weeks, intraperitoneally injected with streptozotocin (STZ) to induce NAFLD, and, then, treated with H8 (3 or 6 mg/kg/day) or curcumin (6 mg/kg/day) for 4 weeks, to evaluate the effects of H8 on NAFLD. H8 significantly alleviated HFD+STZ-induced lipid accumulation, fibrosis, and inflammation as well as improved liver function. Moreover, 11β-HSD1 overexpression was established by transfecting animals and HepG2 cells with lentivirus, carrying the 11β-HSD1 gene, to confirm that H8 improved NAFLD, by reducing 11β-HSD1. An AMP-activated protein kinase (AMPK) inhibitor (Compound C, 10 μM for 2 h) was used to confirm that H8 increased AMPK, by inhibiting 11β-HSD1, thereby restoring lipid metabolic homeostasis. A silencing-related enzyme 1 (SIRT1) inhibitor (EX572, 10 μM for 4 h) and a SIRT1 activator (SRT1720, 1 μM for 4 h) were used to confirm that H8 exerted anti-inflammatory effects, by elevating SIRT1 expression. Our findings demonstrate that H8 alleviates hepatic steatosis, by inhibiting 11β-HSD1, which activates the AMPK/SIRT1 signaling pathway.

## 1. Introduction

Non-alcoholic fatty liver disease (NAFLD) is a chronic liver disease with a gradually increasing incidence worldwide [1,2,3], contributing to an increased risk of metabolic syndrome and cardiovascular disease [2,4]. Although there is no effective treatment for NAFLD, lipid and inflammatory responses are considered key mechanisms in developing the condition. As a result, the regulation of hepatic lipid metabolic homeostasis and anti-inflammatory therapy have received increasing research attention [5,6,7].

In mammals, the AMP-activated protein kinase (AMPK), an energy switch, is activated when Thr 172 of AMPK is phosphorylated. It controls the cell processes of lipid metabolism, by reducing cholesterol and triglyceride synthesis and activating fatty-acid oxidation in the liver [8]. Silencing-related enzyme 1 (SIRT1) has been found to play a beneficial role in regulating hepatic inflammation and lipid metabolism [5]. AMPK activation enhances SIRT1 activity, and they both play important roles in regulating cellular metabolism [9]. Previous research has identified AMPK as a potential target for NAFLD therapy. In addition, 11β-ydroxysteroid dehydrogenase 1 (11β-HSD1) is present in the liver and increases local glucocorticoid levels, by activating 11-dehydrocorticosterone [10]. Glucocorticoids inhibit AMPK activity, by stimulating gluconeogenesis [11]. Therefore, inhibiting 11β-HSD1 to reduce AMPK activity might be a therapeutic target, for treating metabolic disorders in the liver.

Curcumin, a polyphenol component of the traditional medicinal spice turmeric, has demonstrated anti-inflammatory, antioxidant, and anticancer properties as well as exerts control over obesity and diabetes [12]. In addition, curcumin is a natural 11β-HSD1 inhibitor, demonstrated to improve lipid metabolism in rats fed a high-fat diet (HFD) [13]. A recent randomized, controlled trial revealed that curcumin reduces liver fat content, body-mass index, total serum cholesterol (TC), low-density lipoprotein cholesterol (LDL), triglycerides (TG), aspartate aminotransferase (AST), alanine aminotransferase (ALT), glucose (GLU) and glycated hemoglobin levels, in patients with NAFLD [14]. Researchers have, also, demonstrated that 11β-HSD1 inhibition significantly reduces hepatic lipid levels, in patients with NAFLD [15]. Therefore, we hypothesized that curcumin inhibition of 11β-HSD1 may be an effective target for NAFLD treatment. However, the clinical therapeutic efficacy of curcumin is limited, due to its poor absorption, low oral bioavailability, and rapid degradation. We synthesized a curcumin analog: (2E,5E)-2,5-bis [2-fluoro-6-(trifluoromethyl) benzylidene] cyclopentanone (H8). We, previously, reported that H8 reduces visceral fat as well as lowers blood glucose and anti-insulin resistance [16]. However, the specific mechanisms, by which H8 may protect against NAFLD, are unknown.

In the current study, we explored whether H8 regulates lipid metabolism and exerts anti-inflammatory effects, by restoring AMPK/SIRT1 signaling in NAFLD models.

## 2. Materials and Methods

### 2.1. Material

H8 was synthesized and characterized, as previously reported [17]. H8 and curcumin (Sigma, Shanghai, China) were dissolved in 1% sodium carboxymethyl cellulose (CMC-Na) for the in vivo experiments and dissolved in dimethyl sulfoxide (99.7%; Sigma) for the in vitro experiments. Streptozotocin (STZ) (Sigma), Compound C (MCE, Shanghai, China), oleic acid (OA) (Sigma), palmitic acid (PA) (Sigma), SRT1720 (Solarbio, Beijing, China), and EX527 (MCE) were all utilized in this study.

### 2.2. Experimental Animals

Male Sprague Dawley (SD) rats and male C57BL/6J mice were purchased from Liaoning Changsheng Biotechnology Co., Ltd., production license number SCXK (2015-0001; Liaoning, China). All animal studies were conducted in accordance with the ethical standards of the “Guidelines for the Care and Use of Laboratory Animals”, approved by the Mudanjiang Medical University Committee for Animal Experiments (Certificate No. 9 [2020]). The animals were housed in a specific pathogen-free environment, maintained at 22 °C and 65% humidity, with a light/dark cycle of 12 h; rodent nestlets and cardboard homes were provided as environmental enrichment. The animals had free access to standard animal food and water. All the experimental animals were acclimated to the new environment in the animal room, for a week prior to the experiment.

Forty male SD rats were randomly divided into five groups (*n* = 8). The control group was fed a normal diet (3601 kcal/kg, 10% fat, 75.9% carbohydrates, and 14.1% protein, as a percentage of kcal; Trophic Animal Feed High-Tech Co., Ltd., Jiangsu, China). The HFD+STZ group was fed an HFD (5000 kcal/kg, 60% fat, 25.9% carbohydrate, and 14.1% protein, as a percentage of kcal) combined with an intraperitoneal streptozotocin (STZ) injection (25 mg/kg). The low-dose H8 (H8-L), high-dose H8 (H8-H), and curcumin groups were HFD+STZ rats orally administered 3 mg/kg/day H8, 6 mg/kg/day H8, and 6 mg/kg/day curcumin, respectively. Four weeks after treatment, all rats were, additionally, euthanized to collect the blood and liver.

Thirty-two male C57BL/6J mice were randomly divided into four groups (*n* = 8). The solution of lentivirus was centrifuged (500× *g* for 10 min), to remove the cellular debris and purify the lentivirus. The administered virus titer was 10^7^ [18]. The tail veins of 16 mice were injected with lentivirus solution, containing pLVX-EF1α-IRES-ZS green plasmid, and the other half were injected with pLVX-11β-HSD1 green plasmid every 3 days, for 6 consecutive injections of 500 μL. After two weeks, the mice transfected with pLVX-EF1α-IRES-ZS green plasmid were gavaged with CMC-Na (Con group) and H8 (5 mg/kg/day) (Con+H8 group), respectively; the mice transfected with pLVX-11β-HSD1 green plasmid were gavaged with CMC-Na (11β-HSD1 group) and with H8 (5 mg/kg/day) (11β-HSD1+H8), respectively. All mice were, additionally, euthanized at four weeks. Individual body and liver tissue weights were measured. Blood and other tissues were collected and snap frozen in liquid nitrogen for subsequent analyses.

### 2.3. Cell-Culture Treatments and Transfection

HepG2 cells (hepatocellular carcinoma) were purchased from the Procell Life Science Technology Co., Ltd., China, (CL-0103); these cells are useful for constructing the conditions of lipid deposition [19,20]. The cells were cultured in DMEM (Gibco, Carlsbad, CA, USA), comprising 10% fetal bovine serum (FBS) (Gibco) and 1% antibiotic (Invitrogen, Carlsbad, CA, USA), at 37 °C in a humidified 5% CO_2_ atmosphere. For each experiment, HepG2 cells were used in the logarithmic phase. The HepG2 cells (1 × 10^5^ cells/well) were cultivated in 6-well plates, and the cells were adhered to the plate walls for 12 h. To establish the hepatocyte-lipid-accumulation state, the HepG2 cells were incubated with a 1 mM solution of free fatty acids (FFAs), composed of OA and PA (2:1) in a serum-free DMEM medium, for 24 h [21,22]. We tested the biosafety using the MTT assay and found that the concentration (1 mM) did not affect cell activity (Appendix A). FFA-stimulated HepG2 cells were treated with H8-L (2.5 μM), H8-M (5 μM), and H8-H (10 μM), for 24 h. To induce 11β-HSD1 overexpression, we transfected HepG2 cells with pLVX-EF1α-IRES-ZS green plasmid as a Con group or pLVX-11β-HSD1 green plasmid as a 11β-HSD1 group, at 37 °C in a humidified 5% CO_2_ atmosphere, for 24 h. We used fluorescence microscopy to determine transfection efficiency. We determined that the transfection efficiency was higher after 48 h of transfection (Appendix A). Therefore, the cells were transfected for 48 h and treated with H8 (5 μM) for 24 h. To verify the role of AMPK and SIRT1, we pretreated the cells with the AMPK inhibitor Compound C (10 μM) for 2 h or either the SIRT1 activator SRT1720 (1 μM) or the SIRT1 inhibitor EX527 (10 μM) for 4 h. We confirmed the effect of Compound C, SRT1720, and EX527 using Western blotting (Appendix A).

### 2.4. Biochemical Analysis

The biochemical parameters measured in rodent sera included serum cholinesterase (CHE), ALT, AST, TC, TG, GLU, LDL, and high-density lipoprotein (HDL), using an automatic analyzer (Beckman Coulter, Brea, CA, USA) in accordance with the instructions of the manufacturer. A human-glucocorticoid-enzyme-linked immunosorbent assay (ELISA) Kit (SAB Biotech, College Park, MD, USA) was used to establish a standard curve, to detect intracellular glucocorticoid levels. The absorbance at the corresponding maximum absorption wavelength was measured with a spectrophotometer.

### 2.5. Morphological Examination

Liver tissues from the rodents were fixed in 10% formalin for 48 h, dehydrated via an ethanol gradient, and, then, embedded in paraffin wax. The paraffin-embedded tissues were sliced into 4 μm sections, with a microtome (Leica, Wetzlar, Germany). In accordance with the instructions of the manufacturer, liver steatosis and liver fibrosis were measured by hematoxylin and eosin (H&E) as well as Masson staining (Solarbio, Beijing, China). The frozen-compound-embedded tissues were sliced into 10 μm sections and used for Oil red O staining (Sigma), to reveal the presence of fat droplets in the liver. The sections were epimerized with 0.3% Triton X (Solarbio) for 3 min, rinsed with phosphate-buffered saline (PBS), and, then, stained with the Oil red O solution for 45 min at 65 °C. The samples were washed twice with PBS and, then, fixed with 10% formalin for 15 min, to observe the cells. Oil red O staining was performed, as described previously. All slides were mounted with resin or a glycerol-gelatin mix and examined under an optical microscope (Olympus, Tokyo, Japan).

### 2.6. RNA Isolation and Quantitative Real-Time PCR

Total RNA was extracted from the frozen liver tissue (50 mg) or cells (1 × 10^5^ cells/well), using an HP total RNA kit (Omega, China), according to the manufacturer’s instructions. Total RNA (1 μg) was reverse transcribed, using an oligo (dT) 16 primer to generate cDNA, using the Transcriptor cDNA Synth Kit (Roche, Basel, Switzerland). Amplification reactions were conducted on an amplification system (Applied Biosystems, New York, NY, USA). The cDNA and primers were prepared in accordance with the instruction manual of the manufacturer, using the FastStart Universal SYBR Green Master (Roche, Basel, Switzerland); the primers are listed in Appendix A. The amplification conditions were the same as described in the instructions of the manufacturer, and 40 cycles were used for amplification, to measure the mRNA-expression levels in the liver tissue or cells. The results derived from the RT-PCR data were statistically analyzed, using the change in Ct values, and normalized by GAPDH Ct values.

### 2.7. Western Blotting

The liver tissue or cells were lysed with RIPA buffer (Solarbio) as well as the protease and phosphatase-inhibitor mixture ABC (HaiGene, Heilongjiang, China). The protein concentrations were measured, using a Nanodrop 2000 (Thermo Fisher, Shanghai, China). Equal amounts of protein samples (50 μg) were separated on 10% SDS-PAGE and electrophoretically transferred to polyvinylidene difluoride (PVDF), to cover the primary antibodies for 12 h at 4 °C; the primary antibodies are provided in Appendix A. The PVDF membranes were washed three times with Western washing buffer the next day and incubated with horseradish peroxidase-conjugated secondary antibody (ZSGB-BIO, Beijing, China), for 45 min at room temperature. An ECL Western Blotting Substrate kit (Solarbio) was used to detect the chromogen. Finally, the positive bands were visualized, with the enhanced chemiluminescence system (Merck Millipore, Billerica, MA, USA), as well as quantified and normalized with β-actin (Cell Signaling, 1:1000), using ImageJ software (Image J 1.8.0).

### 2.8. Immunocytochemistry

Cells were prepared in a 6-well plate and fixed with 10% formalin, for 1 h. The fixed cells were washed three times with PBS and permeabilized with 0.3% Triton X-100, for 30 s. They were washed four times with PBS and blocked with 10% normal goat serum (ZSGB-BIO, Beijing, China) for 45 min. After covering with primary p-AMPK (1: 500, Abcam, Cambridge, MA, USA) and SIRT1 (1: 250, Affinity, Jiangsu, China), for 12 h at 4 °C, the cells were incubated with the secondary antibody IgG (H&L), conjugated with FITC (1: 250, Affinity), and the IgG secondary antibody (H&L), conjugated with CY3 (1: 250, Affinity), for 45 min. The cell nuclei were stained with DAPI (Yeasen, Shanghai, China). Then, the fluorescence photographs were acquired, using a confocal microscope (Olympus, Tokyo, Japan).

### 2.9. Statistical Analysis

The data were assessed by GraphPad Prism version 8.2.1 software (GraphPad Software Inc., San Diego, CA, USA), using a one-way analysis of variance (ANOVA) for comparisons, followed by Bartlett’s test for equal variances. Then, the data were analyzed using Tukey’s test. All data pairs were compared, as appropriate. The statistical analysis was used to assess datasets containing groups of n ≥ 3, in which n represents the number of separate experiments and the number of rodents. All the data are presented as the mean ± standard error of the mean (SEM). A *p*-value ≤ 0.05 was considered statistically significant.

## 3. Results

### 3.1. H8 Alleviates NAFLD in Rats

HFD+STZ-group rats had lower body weight and a higher liver weight/body weight ratio than control-group rats (*p* < 0.01), the changes in body weight are shown in Appendix A. Although the body weight of rats in the H8-treated and curcumin-treated groups was not significantly different from the HFD+STZ group, the H8 treatment almost completely normalized the liver weight and liver weight/body weight ratio, to those of the control (*p* < 0.01) (Figure 1A–C).

In addition, H&E staining indicated that the HFD+STZ group showed significant hepatic vacuolation and hepatocyte steatosis, compared to the control group. In contrast, H8 treatment significantly alleviated hepatic vacuolation and hepatocyte damage in HFD+STZ rats, and the protective effect was stronger than that of curcumin (Figure 1E, upper panel). Masson staining indicated that the HFD+STZ group had a higher degree of fibrosis around the central hepatic vein vessels than the control group. The H8-H treatment decreased liver fibrosis more obviously than curcumin and H8-L treatments (Figure 1E, middle panel). The Oil red O staining results, also, showed increased lipid droplet size and lipid accumulation in liver tissues of the HFD+STZ group, compared with the control, which was more obviously attenuated by the H8-H treatment than the curcumin and H8-L treatments (Figure 1E, lower panel) (see Appendix A, for oil-red grayscale values).

The HFD+STZ group had significantly higher serum AST, ALT, AST/ALT ratio, CHE, TC, TG, GLU, and LDL levels as well as reduced HDL levels than the control (*p* < 0.05 or *p* < 0.01), reflecting a global metabolic disorder and an impairment in liver function. Notably, all these changes were significantly reversed, by treatment with both H8 and curcumin (*p* < 0.05 or *p* < 0.01) (Figure 1D,F–M). Among them, the effect of H8 on AST/ALT, TC, and GLU was superior to that of curcumin.

The above results showed that rats in the HFD+STZ group developed obvious hepatic steatosis with NAFLD characteristics, while H8 alleviates this phenomenon.

### 3.2. H8 Improves Lipid Metabolism in NAFLD Models

Compared to the control, the HFD+STZ group had reduced CPT-1β, HSL, and p-ACC as well as increased SREBP1 levels. The above effects were reversed by treatment of H8 and curcumin (*p* < 0.01); however, curcumin did not cause changes in the HSL protein levels (Figure 2A–C). In the cell experiments, we, first, confirmed by MTT assay that cytotoxicity was not significant, when cells were treated with H8 at concentrations of 2.5–10 μM, for 24 h. The results are shown in Appendix A. Moreover, we found that the FFAs-induced-HepG2-lipid-accumulation model had reduced CPT-1β, HSL, and p-ACC expression as well as elevated SREBP1 and FASN expression, compared to the control. Notably, these effects were reversed, when the H8 concentration was 5 μM (*p* < 0.05 or *p* < 0.01). Consistent with the results of animal experiments, H8 did not affect FASN at the genetic level (Figure 2D–F).

The above results demonstrate that H8 promoted lipid catabolism and inhibited anabolism in hepatocytes with NAFLD.

### 3.3. H8 Alleviates Liver Injury by Inhibiting 11β-HSD1

HFD+STZ rats had significantly elevated 11β-HSD1 but reduced p-AMPK and SIRT1 levels, compared to the controls. H8 effectively reversed these results (*p* < 0.01) (Figure 3A–C). In HepG2, 11β-HSD1 was similarly reduced in response to H8, while SIRT1 and p-AMPK levels were increased (*p* < 0.01) (Figure 3D–F). Immunofluorescence-staining results further confirmed that H8 restored the expression of p-AMPK and SIRT1, in FFAs-induced HepG2 cells (Figure 3G). The above results showed that the level of 11β-HSD1 increased, but the levels of P-AMPK and SIRT1 decreased, under high-lipid conditions, while H8 reverses these phenomena.

To determine if H8 inhibition of 11β-HSD1 improves liver function, we established a stable expression system of 11β-HSD1 in mice and cells.

Body weight was not significantly changed, but liver weight and the liver weight/body weight ratio increased, in mice injected with pLVX-11β-HSD1. Treatment with H8 significantly restored liver weight and liver weight/body weight to normal, in the 11β-HSD1 group (*p* < 0.05) (Figure 4A–C). Liver and cell-lipid-droplet changes in the different groups, defined by Oil red O staining (Figure 4D,E), were increased in the groups transfected with 11β-HSD1 lentiviral plasmids and decreased by H8 treatment. The 11β-HSD1-group mice had elevated serum ratios of AST/ALT, CHE, TC, TG, and GLU (*p* < 0.01 or *p* < 0.05), reflecting the occurrence of liver injury. However, all the above damage was alleviated, after treatment with H8 (Figure 4F–J).

The above results indicated that H8 ameliorates liver injury, probably through the inhibition of 11β-HSD1.

### 3.4. H8 Elevates AMPK/SIRT1 by Inhibiting 11β-HSD1

We found, in the mice and cells of the 11β-HSD1 group, that treatment with H8 decreased the expression level of 11β-HSD1 (*p* < 0.01 or *p* < 0.05), with elevated expression of SIRT1 and p-AMPK (*p* < 0.01) (Figure 5), indicating that H8 promotes the APMK/SIRT1 pathway, by inhibiting 11β-HSD1. ELISA results show that glucocorticoid levels were elevated in cells transfected with 11β-HSD1. Furthermore, after H8 treatment, glucocorticoid levels decreased (*p* < 0.01) (Figure 5D).

We used the AMPK inhibitor (Compound C) and SIRT1 inhibitor (EX527), to investigate the interaction of 11β-HSD1 with AMPK and SIRT1.

In HepG2, Compound C significantly reduced SIRT1 and elevated 11β-HSD1 protein-expression levels (Figure 6A–C); EX527 significantly decreased p-AMPK and elevated 11β-HSD1 protein-expression levels (Figure 6D–F). Treatment of H8 restored the expression levels of AMPK and SIRT1, thus decreasing the protein expression of 11β-HSD1 (*p* < 0.05 or *p* < 0.01) (Figure 6). This suggests that the expression level of 11β-HSD1 is negatively correlated with the AMPK/SIRT1 signaling pathway.

### 3.5. H8 Improves Hepatocyte Lipid Metabolic Disorder by Promoting AMPK

We used the AMPK inhibitor (Compound C) to investigate whether H8 regulates lipid metabolism in hepatocytes, through AMPK.

Oil red O staining (Figure 7A) showed an increase in lipid-droplet expression in the Compound C and FFAs groups, suggesting that inhibition of AMPK increased lipid synthesis equally, regardless of high lipid conditions, while H8 significantly reduced lipid-droplet accumulation, suggesting the potential effect of balancing lipid metabolism.

The Compound C group demonstrated lower CPT-1β, HSL, and p-ACC as well as higher SREBP1 and FASN than the Con group (*p* < 0.05 or *p* < 0.01); these results were consistent with the FFAs group. In contrast, H8 had a significant recovery effect on lipid-metabolism disorder in the Compound C group (*p* < 0.01 or *p* < 0.05) (Figure 7A–C), suggesting that H8 balances lipid metabolism, by restoring AMPK level.

### 3.6. H8 Exerts Anti-Inflammatory Effects by Activation of SIRT1

PPAR-γ [23] and PGC-1α [24], previously reported to have anti-inflammatory effects [25], were used as anti-inflammatory markers in this study.

The inflammatory response was reflected by increased p-p65, IL-6, and TNF-α as well as decreased levels of PPAR-γ and PGC-1α in HFD+STZ groups, compared to the control group. Moreover, treatment with H8 prevented an inflammatory response in HFD+STZ rats (*p* < 0.05 or *p* < 0.01) (Figure 8A–C).

Consistent with animal experiments, H8 effectively decreased TNF-α and elevated PGC-1α (*p* < 0.01) at the gene level, in HepG2 cells (Figure 9A). The EX527 group had elevated IL-6 and decreased PPAR-γ, while the SRT1720 group had decreased IL-17 and IL-6 as well as increased PGC-1α (*p* < 0.05 or *p* < 0.01), indicating that inhibited SIRT1 promotes inflammatory responses. Although the results were not statistically significant, the p-p65 level tended to increase in the EX527 group and decrease in the SRT1720 group. In contrast, the H8-treatment group reduced (*p* < 0.05) the level of p-p65 (Figure 9B,C).

Our study suggests that the restoration of SIRT1 is a potential pathway for H8 to exert anti-inflammatory effects.

## 4. Discussion

Our study found that the livers of HFD+STZ induced rats have NAFLD characteristics, while H8 improved the symptoms of NAFLD and is more effective than curcumin.

We used the NAFLD rat model of hepatic steatosis, feeding the rats an HFD and intraperitoneally injecting them with STZ [26]. The rats in the HFD+STZ group began to lose weight after STZ injection and, ultimately, weighed less than the control group (Appendix A), exhibiting the typical diabetic symptoms of polyuria, polydipsia, and weight loss. H&E staining and Oil red O staining showed that the HFD+STZ group had developed liver steatosis, with NAFLD characteristics. H8 restored liver histology and serum biochemical indexes significantly, suggesting that it improves liver lipid metabolism.

Improving hepatic lipid metabolism is significant, when treating NAFLD induced by lipid accumulation or obesity. To elucidate how H8 balances lipid metabolism, we tested indicators related to lipid metabolism in the rat liver. Phosphorylation of Ser79 in ACC reduces malonyl coenzyme A levels [27,28]. Decreased malonyl coenzyme A increases the level of CPT-1β, to restore fatty acid β-oxidation [29]. SREBP1 regulates the transcriptional activation of the lipid-synthesis-rate-limiting enzyme FASN, which is involved in cholesterol uptake as well as generation of fatty acids and lipids. Furthermore, it is associated with increased lipogenesis in NAFLD [30]. H8 increased the phosphorylation level of ACC and inhibited the level of SREBP, suggesting a role in the hepaticlipid-metabolism-balancing effect. However, its target of effect requires further study.

It has been reported that 11β-HSD1 has important effects on lipid metabolism, by activating glucocorticoids in peripheral tissues [31]. Curcumin is a natural inhibitor of 11β-HSD1, extracted from the Chinese medicinal spice turmeric, which demonstrates multiple physiological functions [13] and low bioavailability [32]. We constructed a curcumin analog, H8, that stably and strongly inhibited 11β-HSD1 [17], alleviated insulin resistance, AND reduced glucocorticoid levels by inhibiting 11β-HSD1. We demonstrated that elevated 11β-HSD1 causes liver injury, increases lipid accumulation, and decreases AMPK and SIRT1 activities. In contrast, treatment with H8 reverses these effects, by inhibiting 11β-HSD1. This function may work by regulating glucocorticoid levels.

A previous study showed that glucocorticoid-induced kinases directly inhibit AMPK [33], which may explain how glucocorticoid alterations affect lipid metabolism. In mammalian cells, AMPK activation promotes Ser79 phosphorylation in ACC. Moreover, AMPK activation enhances SREBP1 phosphorylation at Ser 372, to inhibit SREBP1 maturation. Recently published studies have demonstrated that acute glucocorticoid exposure increases cardiac-fatty-acid oxidation, by activating AMPK [34]. In contrast, in mice chronically exposed to glucocorticoids, AMPK phosphorylation levels and lipid metabolism gene levels have been shown to decrease with activation of 11β-HSD1 [19,35,36]. These phenomena may be due to AMPK inhibition associated with the 11β-HSD1-induced time-dependence of cortisol. In addition, it has been shown that AMPK increases the organismal level of NADH, which activates SIRT1 expression [37]. Conversely, SIRT1 promotes LKB1-mediated AMPK activation [38], to attenuate lipid accumulation by regulating glucose tolerance [39] and inducing white-adipose-tissue browning [40]. Based on the above studies, we propose that H8 treats NAFLD, by restoring AMPK and SIRT1 through inhibition of 11β-HSD1. We overexpressed 11β-HSD1 in mice and inhibited AMPK or SIRT1 in HepG2 cells, confirming that 11β-HSD1 expression is inversely correlated to the expression of AMPK/SIRT1 pathway.

To confirm the role of AMPK in the treatment of lipid metabolism disorders in H8, we used compound C [41] to inhibit AMPK activity and observe lipid metabolism indicators in HepG2 cells, when AMPK activity was inhibited. Our experiments showed that H8 increases the expression of proteins related to lipid catabolism and inhibits the expression of lipid-synthesis proteins, to balance lipid metabolism by restoring AMPK levels. H8 does not affect the gene expression level of FASN, suggesting that H8 regulates FASN at the post-transcriptional level. These results imply that the function of H8, in alleviating disorders of hepatic lipid metabolism, is AMPK dependent.

It has been shown that inhibition of 11β-HSD1 effectively reduces the gene-expression levels of pro-inflammatory factors, suggesting that inhibition of 11β-HSD1 may have anti-inflammatory effects, independent of lipid metabolism [42]. We suggest that this effect may be related to SIRT1, which exerts anti-inflammatory effects by activating PPAR-γ to promote macrophage polarization to the M2 phenotype [43]. SIRT1, also, indirectly reduces the activation of the NF-κB signaling pathway and pro-inflammatory cytokines, such as TNF-α and IL-6 [37,44], by activating PGC-1α [45].

To confirm that H8 exerts an anti-inflammatory effect, by affecting the level of SIRT1, we used SIRT1 inhibitor EX527 and activator SRT172 to investigate the role of SIRT1 in the process [46]. First, our results confirmed that H8 effectively reduces the expression level of the inflammatory factor in the liver of HFD+STZ rats. In addition, we found higher levels of the TNF-α gene, p-p65, and IL-17 protein expression in HepG2 cells, which may result from HepG2 being a hepatocellular carcinoma cell, with inherently high levels of inflammation. Our results, also, revealed that overexpression of SIRT1 effectively attenuates the expression of inflammatory factors, and inhibition of SIRT1 eliminates the anti-inflammatory properties of H8, possibly because H8 can only indirectly activate SIRT1 through AMPK.

Notably, besides being a downstream target of SIRT1, PPAR-γ and PGC-1α are closely associated with regulating glycolipid metabolism. PPAR-γ mitigates NAFLD progression, by regulating adipocytokine expression, preventing insulin resistance [47], and stimulating CPT-1β activity, to promote fatty-acid oxidation [48]. A bioinformatics analysis showed that the level of PPAR-γ was reduced under lipid accumulation conditions [19], and that the activation of PPAR-γ reduced TG levels in patients with NAFLD [49]. Whether H8 plays a regulatory role in NAFLD, by affecting glycolipid metabolism through modulation of SIRT1, and an anticancer role, by reducing TNF-α levels, warrants further investigation.

## 5. Conclusions

Our experimental results confirm that H8 balances lipid metabolism and exerts anti-inflammatory effects, by inhibiting 11β-HSD1 and upregulating the AMPK/SIRT1 signaling pathway, providing promising data to inform pharmacological studies of NAFLD treatment.

## Figures and Tables

**Figure 1 nutrients-14-02358-f001:**
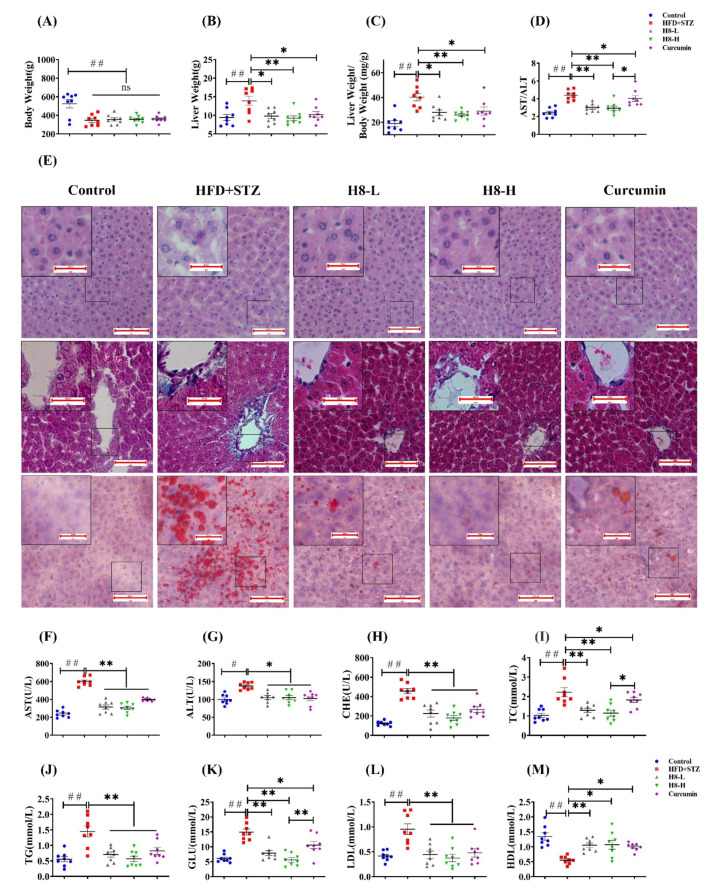
H8 reduces lipid accumulation and improves liver function in HFD+STZ rats. (**A**) Body weight; (**B**) liver weight; (**C**) liver weight/body weight; (**D**) AST/ALT levels. In rat liver: (**E**) H&E staining in upper panel; Masson staining in middle panel; Oil red O staining in lower panel (scale bar, 20 μm or 100 μm). (**F**–**M**) Serum biochemical indexes of rat. The data represent the mean ± SEM. *n* = 8, # *p* < 0.05, ## *p* < 0.01 versus the control; * *p* < 0.05, ** *p* < 0.01 versus the HFD+STZ; ns *p* > 0.05 not significant.

**Figure 2 nutrients-14-02358-f002:**
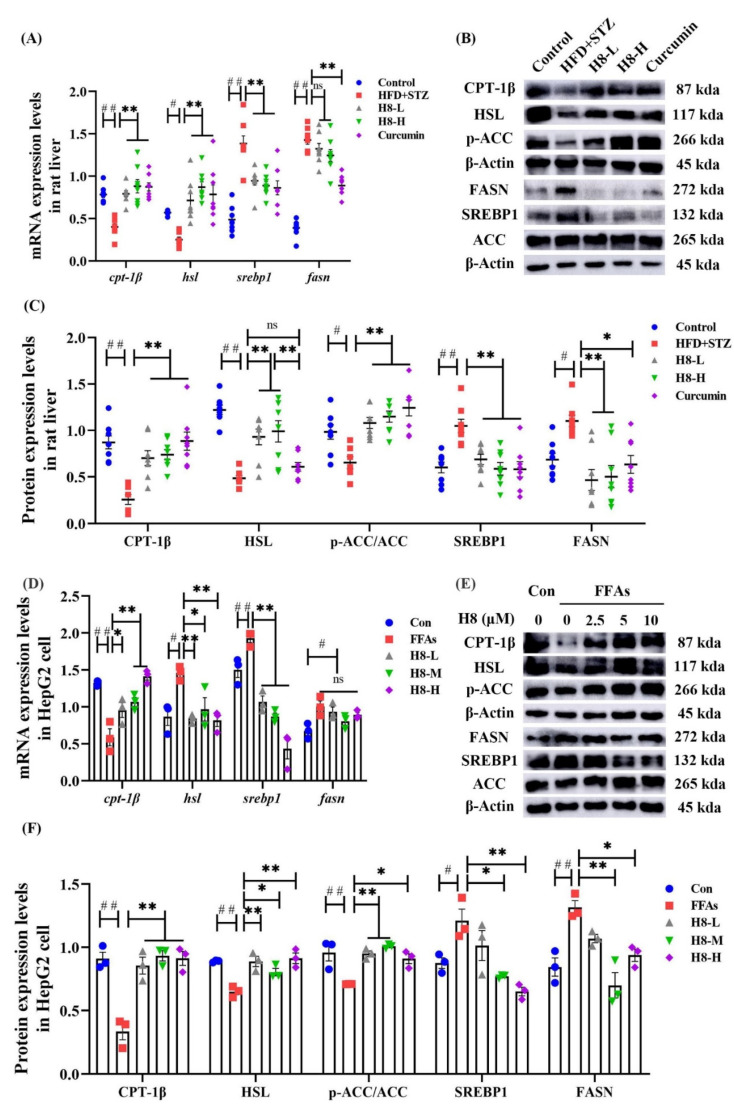
H8 improves the balance of lipid metabolism in hepatocytes. In rat liver: (**A**) the relative mRNA expression; (**B**) the representative Western blot images; (**C**) the protein-expression levels. In cells: (**D**) the relative mRNA expression; (**E**) the representative Western blot images; (**F**) the protein-expression levels. The data represent the mean ± SEM. # *p* < 0.05, ## *p* < 0.01 versus the control group; * *p* < 0.05, ** *p* < 0.01 versus the HFD+STZ group; ns *p* > 0.05 not significant.

**Figure 3 nutrients-14-02358-f003:**
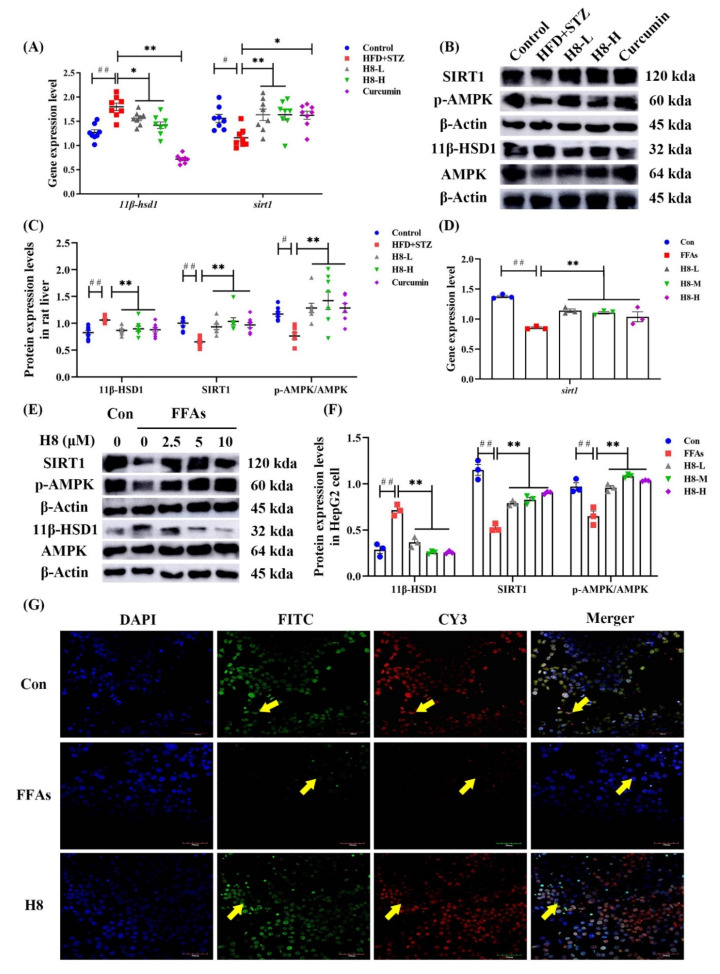
H8 inhibits 11β-HSD1 as well as increases the expression of SIRT1 and phosphorylation of AMPK. In rat liver: (**A**) the relative mRNA expression; (**B**) the representative Western blot images; (**C**) the protein=expression levels. In cells: (**D**) the relative mRNA-expression levels; (**E**) the representative Western blot images; (**F**) the protein expression levels; (**G**) immunofluorescence staining, green fluorescence FITC represents the protein P-AMPK, red fluorescence CY3 represents the protein SIRT1, nuclei were visualized by DAPI (scale bar, 50 μm). The data represent the mean ± SEM. # *p* < 0.05, ## *p* < 0.01 versus the control group; * *p* < 0.05, ** *p* < 0.01 versus the HFD+STZ group.

**Figure 4 nutrients-14-02358-f004:**
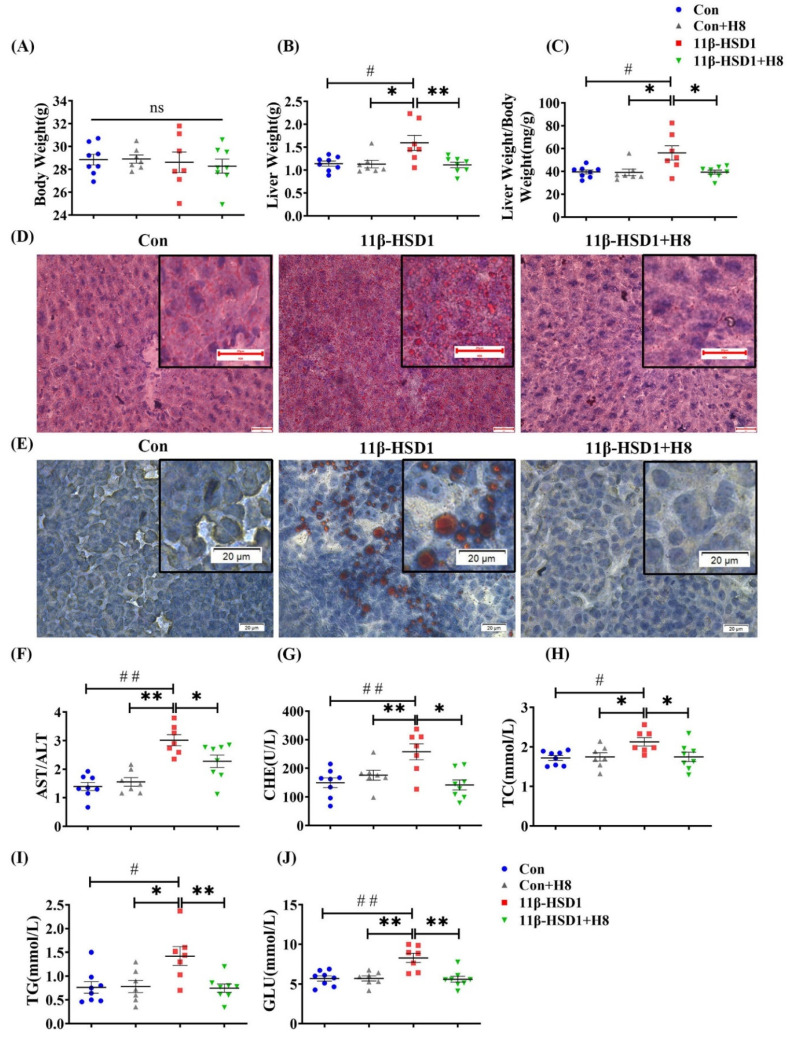
H8 alleviates liver injury, by inhibiting 11β-HSD1. In mice: (**A**) body weight; (**B**) liver weight; (**C**) liver weight/body weight ratios. Oil red O staining images of (**D**) mouse liver (scale bar, 20 μm); (**E**) HepG2 cells (scale bar, 20 μm). (**F**–**J**) Serum biochemical indexes of mice. The data represent the mean ± SEM. # *p* < 0.05, ## *p* < 0.01 versus the Con group; * *p* < 0.05 and ** *p* < 0.01 versus the 11β-HSD1 group; ns *p* > 0.05 not significant.

**Figure 5 nutrients-14-02358-f005:**
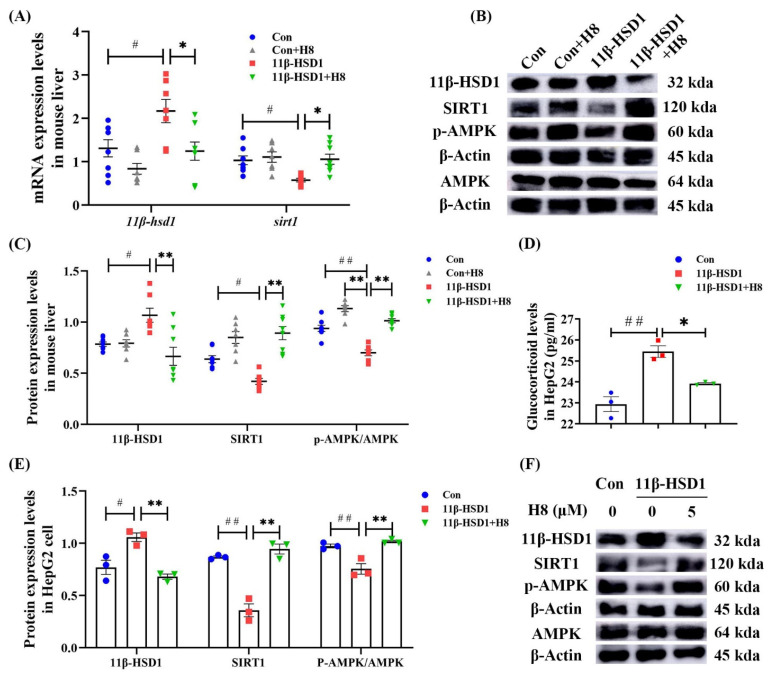
H8 elevates AMPK/SIRT1, by inhibiting 11β-HSD1. In mouse liver: (**A**) the relative mRNA expression; (**B**) the representative Western blot images; (**C**) the protein-expression levels. In cells: (**D**) glucocorticoid levels; (**E**) the protein-expression levels; (**F**) the representative Western blot images. The data represent the mean ± SEM. # *p* < 0.05, ## *p* < 0.01 versus the Con group; * *p* < 0.05, ** *p* < 0.01 versus the 11β-HSD1 group.

**Figure 6 nutrients-14-02358-f006:**
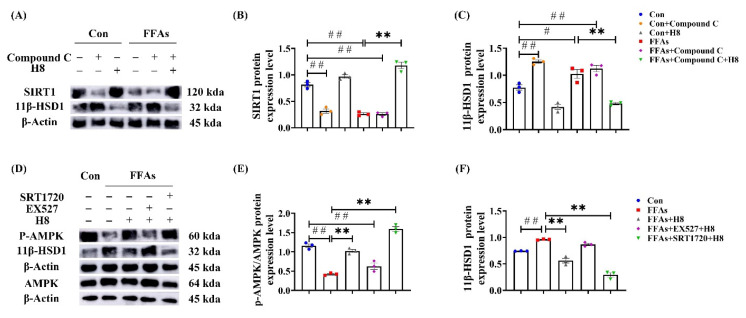
AMPK/SIRT1 signaling pathway negatively regulates 11β-HSD1. In HepG2: (**A**) the representative Western blot images; (**B**) SIRT1 protein-expression level; (**C**) 11β-HSD1 protein-expression level; (**D**) the representative Western blot images; (**E**) p-AMPK protein-expression level; (**F**) 11β-HSD1 protein-expression level. The data represent the mean ± SEM. *n* = 3, # *p* < 0.05, ## *p* < 0.01 versus the Con group; ** *p* < 0.01 versus the FFAs group.

**Figure 7 nutrients-14-02358-f007:**
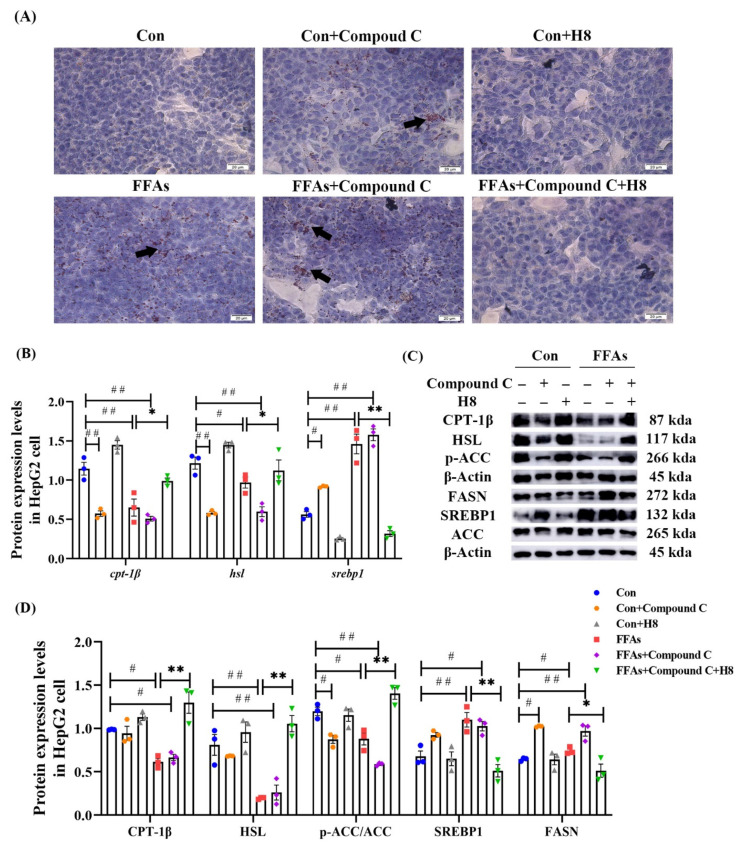
H8 restores the abnormal lipid metabolism, caused by the inhibition of AMPK. In HepG2 cells: (**A**) Oil red O staining images (scale bar, 20 μm); (**B**) the relative mRNA expression; (**C**) the representative Western blot images; (**D**) the protein expression levels. The data represent the mean ± SEM. *n* = 3, # *p* < 0.05, ## *p* < 0.01 versus the Con group; * *p* < 0.05, ** *p* < 0.01 versus the FFAs group.

**Figure 8 nutrients-14-02358-f008:**
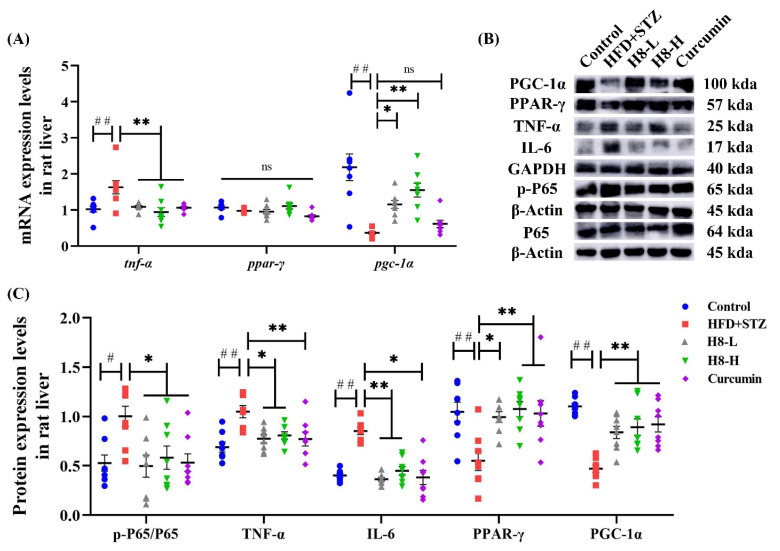
H8 alleviates the inflammatory response in HFD+STZ rat liver. In rat liver: (**A**) the relative mRNA expression; (**B**) the representative Western blot images; (**C**) the protein-expression levels. The data represent the mean ± SEM. *n* = 8, # *p* < 0.05, ## *p* < 0.01 versus the control; * *p* < 0.05, ** *p* < 0.01 versus the HFD+STZ; ns *p* > 0.05 not significant.

**Figure 9 nutrients-14-02358-f009:**
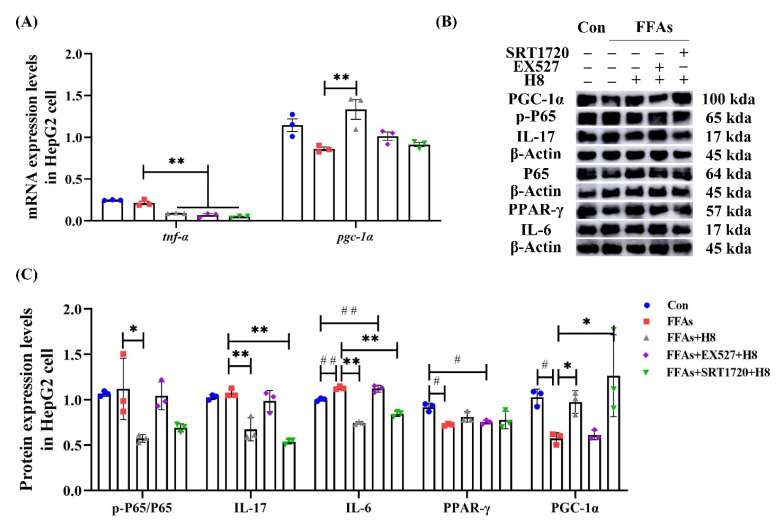
H8 exerts anti-inflammatory effects by activating SIRT1. In cells: (**A**) the relative mRNA expression; (**B**) the representative Western blot images; (**C**) the protein-expression levels. The data represent the mean ± SEM. *n* = 3, # *p* < 0.05, ## *p* < 0.01 versus the Con group; * *p* < 0.05, ** *p* < 0.01 versus the FFAs group.

## Data Availability

Not applicable. The data presented in this study are available in the Appendix A.

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
