# Peer review of "11β-HSD1 Inhibitor Alleviates Non-Alcoholic Fatty Liver Disease by Activating the AMPK/SIRT1 Signaling Pathway"

_nutrients, 2022, doi:10.3390/nu14112358_

Round 1
Reviewer 1 Report
Chen et al have presented a very interesting data about the mechanism of 11β-HSD1 inhibitor (H8) on high-fat in vivo and in vitro.
Unfortunately, the manuscript needs extensive work.Â
My main concern is the writing of the manuscript. Results and Discussion are not clearly presented. They should provide the reader with detailed information and, more importantly, take the reader step by step through the development of the work. I strongly suggest major and extensive rewriting of the above chapters, including figure legends.Â
Reviewer 2 Report
The authors presented interesting findings that move us a step closer to managing and understanding potential targets in NAFLD. However, I have some comments to be taken into consideration.
- Firstly, in the introduction, the authors should highlight the role of curcumin and whether it was studied before in NAFLD; they clearly mention its benefits in relation to diabetes and obesity but highlighting its role in NAFLD is more relevant and justifies why they choose to investigate H8 its synthetic derivative, and why 11β-HSD1 inhibition is a potential target to manage NAFLD. I have gone through the review article the authors referenced to claim curcumin is a natural 11β-HSD1 inhibitor (reference 12), it does not clearly discuss literature supporting their claim thus a stronger reference should be used.
Suggested reference:
Hu, G. X., Lin, H., Lian, Q. Q., Zhou, S. H., Guo, J., Zhou, H. Y., Chu, Y., & Ge, R. S. (2013). Curcumin as a potent and selective inhibitor of 11β-hydroxysteroid dehydrogenase 1: improving lipid profiles in high-fat-diet-treated rats. PloS one, 8(3), e49976. https://doi.org/10.1371/journal.pone.0049976
- Secondly, in the methods section under the experimental animals’ description, I believe it needs minor rephrasing to be more understandable to the audience. For instance, the second paragraph should mention clearly that the NAFLD groups had one control group (HFD, STZ) and three treatment groups. I believe this will make it easier to identify the 5 groups without confusion.
Reviewer 3 Report
The work of Chen et al. entitled: ’11b-HSD1 inhibitor alleviates non-alcoholic fatty liver disease by activating the AMPK/SIRT1 signaling pathway’ evaluated the effect and mechanism of the 11b-HSD1 inhibitor H8 on HFD-induced hepatic steatosis in rats. Animals were fed with HFD for 8 weeks prior to the treatment with H8. The authors could show that the treatment with H8 had beneficial effects on liver function by directly reducing 11b-HSD1. They could further identify the AMPK/SIRT1 signaling pathway as target through which H8 restores lipid metabolic homeostasis.
Â
Â
Overall, the manuscript is well written and structured. However, major points of criticism remained after reading the manuscript.
Â
- How do the authors explain that HFD fed animals have a lower bodyweight than controls? In line with this the presented H&E and ORO-stained images do not show obvious lipid droplet formation and steatosis in the liver which raised the question whether the model worked.
- Especially in figure 4 it looks like there is massive bleeding in the liver. How do the authors explain this finding?
- In the displayed Westerblot analysis the control line does not seem to be representative for the actual blot which raises concerns about the accuracy of the performed analysis.
- The first line of the discussion section states: ’Our study found that the livers of HFD rats have NAFLD characteristics (…)’. This is not supported by the data shown by the authors. Neither do the animals show weight increase nor does the liver histology look like one of a liver with NAFLD after HFD feeding.
Round 2
Reviewer 1 Report
Dear Authors,
Thanks for the revised manuscript.
I do not have further comments.
Reviewer 3 Report
The authors addressed all raised comments and improved the manuscript.
This manuscript is a resubmission of an earlier submission. The following is a list of the peer review reports and author responses from that submission.